# Selected Correlates of Attitudes towards Rape Victims among Polish Medical Students

**DOI:** 10.3390/ijerph19105896

**Published:** 2022-05-12

**Authors:** Lidia Perenc, Justyna Podgórska-Bednarz, Agnieszka Guzik, Mariusz Drużbicki

**Affiliations:** Institute of Health Sciences, University of Rzeszow, 35-310 Rzeszow, Poland; jpodgorska@ur.edu.pl (J.P.-B.); aguzik@ur.edu.pl (A.G.); mdruzbicki@ur.edu.pl (M.D.)

**Keywords:** rape victims, attitudes, medical students

## Abstract

Sexual violence against women, including rape, is a serious public health issue in many countries. Rape victims often meet health professionals in medical institutions for a range of health problems. The aim of this research was investigation of attitudes towards rape victims among medical students. Methods: The study sample consisted of 1183 university students who represented various medical disciplines. The average age of the respondents was 23.3 years. The Attitudes toward Rape Victims Scale (ARVS) was used in this study. Results: Higher scores in men indicate that they held less sympathetic attitudes towards rape victims than women (61.6 vs. 52.6, *p* = 0.0000). Given the univariate interaction, social environment, and religious commitment did not significantly differentiate the respondents in this respect. Students of the medical faculty obtained the lowest results (medicine 49.7 vs. midwifery and nursing: 54.1, other fields: 54.4, *p* = 0.0008), showing much understanding and empathy for rape victims. Conclusions: The surveyed medical students presented moderately positive attitudes towards rape victims, among them men somewhat negative than women who made more pro-victim judgments. Among all medical field of study, medicine was distinguished by higher empathy. Religion and social environment independently do not differentiate respondents in this respect.

## 1. Introduction

Sexual violence against women, especially rape, is a serious public health issue. This form of violence may affect any woman from early childhood to old age. As a result, women who are victimized suffer a range of health problems and their functioning in daily life can be significantly restricted. This is why rape victims often meet health professionals in medical institutions. Statistical data related to rape and other forms of sexual assault that have recently occurred recently in different countries are easily accessible. However, because of inconsistent definitions of rape and different reporting methods, the rape statistics are often misleading and unreliable. This is because rape cases are often not reported by victims. This phenomenon occurs in Poland, as well as in other countries [1,2,3]. According to police statistics, in the 21st century, the number of rapes in Poland was the highest in 2002 (2345 cases) and remained at this level for three consecutive years. Since 2005, we observed a systematic decrease in this number, and in 2017 there were 1262 cases of rape [1]. The observed trend makes Poland, along with Greece and Portugal, one of the European countries with a relatively low number of rapes per 100,000 inhabitants [4]. The countries with the highest rates of rape include Sweden and the United States [5]. However, such statistics should be approached with caution, as various comparisons show that only 2 to 11% of all sexual offenses are disclosed and prosecuted [6,7]. In Poland, there are no reliable data which would indicate what percentage of rapes ends in an unwanted pregnancy. For example, in the USA, the national rate of rape-related pregnancies is estimated at 5% of rape victims for women aged 12 to 45 years [8]. Rapes committed during war are used as a weapon in armed conflicts and contribute to a change of the ethnic structure [9]. The acceptance of rape myths in the community is associated with stigma and trauma-related mental illness of sexual violence survivors [10]. In these circumstances, research conducted on the possibility of shaping empathetic attitudes towards rape victims is of great importance [11].

The authors of this article use the definition of attitude used in psychology and medicine, according to which attitude is a set of emotions, beliefs, and behaviors toward a particular object, person, thing, or event. Attitudes are often the result of personal experience or education, and can have a powerful influence over behavior. As is known, attitudes often are enduring, they can also change under influence of certain factors such as personal experiences or persuasion of significant persons [12].

The present study is aimed at examination of attitudes towards rape victims in a large group of Polish medical students. The authors assume that it is an important, however under-researched problem, especially when we consider the fact that medical personnel play a crucial role in therapy of rape victims. Victims are dependent upon physicians and other medical personnel to gather medical legal evidence if the rape is to be reported and investigated. Moreover, whether they report or not, victims are dependent upon physicians for evaluation and treatment of medical problems which result from sexual assaults. Unfortunately, medical schools often did not adequately train their students in rape examination procedures and attitudes towards this special group of patients. The importance of this study is also due to the fact that, unit now in Poland, unlike other countries, no research has been conducted on the attitudes of medical students towards the rape victims.

The purpose of this study was to assess and understand the variation in the attitudes of medical students towards victims of rape. In particular, the authors wanted to determine the relationship between some basic socio demographic variables and these attitudes.

## 2. Methods

### 2.1. Population

This was a cross-sectional study conducted at the College of Medical Sciences, belonging to the University of Rzeszow, during the period of March 2019–February 2020. At the University of Rzeszow, the number of all students in medical faculties as of 31 December, 2019 was 2785 (women constituted 79.9%—2227 persons), and as of 31 December, 2020–2907 (women constituted 79.1%—2301 persons). The permission from the relevant university authorities was obtained before actual collection of data. The study sample consisted of 1183 university students who represented various medical disciplines such as physiotherapy, nursing, midwifery, dietetics, public health, medicine (future physicians), and emergency medical services. They were students of the 1st, 2nd and 3rdyear of undergraduate studies (or 1st, 2nd and 3rd year of uniform studies) and 1st and 2nd year of graduate studies (or 4th, 5th and 6th year of uniform studies). The surveyed sample was predominantly female, with 1016 subjects (85.9%), with only 167 men (14.1%). Gender inequality also occurred in a similar study conducted by other researchers [13]. In medical studies at the University of Rzeszow, the disproportion between the genders occurs.

The average age of the respondents was 23.3 ± 5.3 years (the average age of women was 23.4 ± 5.6 years, and men—22.7 ± 3.2 years). The remaining demographic characteristics are shown in Table 1.

### 2.2. Procedures and Data Analyses

The Attitudes toward Rape Victims Scale (ARVS) developed by Colleen Ward was used in this study [14]. This tool is used to assess attitudes related to rape victims, with particular emphasis on issues such as credibility, slander, trivialization, blame or deserving punishment. The scale questionnaire consists of 25 items, 17 of which are negative and 8 are positive about rape victims. Individual items are scored on a 5-point Likert scale (direct score: strongly disagree—1, disagree—2, neither agree nor disagree—3, agree—4, strongly agree—5, reversed score (R): strongly disagree—5, disagree—4, neither agree nor disagree—3, agree—2, strongly agree—1), and the final score is the sum of all points. It may range from 25 to 125 points, the average score is 75, the lower quartile—50, the upper quartile—100 points. High results indicate a negative attitude of the examined person towards rape victims. The analysis of the psychometric properties of the ARVS scale showed its high validity (Cronbach’s alpha = 0.83). The greater the number of points, the lower the overall level of empathy towards the rape victim (total score), the more likely, according to the respondent, that rape victim “deserved” the experience of rape (measure is the sum of items: 6, 9, 11, 13, 24, 25); the less reliable according to the respondent, that the situation that took place was actually a rape (victim credibility, measure is the sum of items: 2, 8, 14, 16, 17, 21, 22R); the more the victim is complicit for the rape (victim blame, measure is the sum of items: 3R, 5R, 7R, 10R, 12R, 15R, 19R), the more the victim deserves social exclusion because of the experience of rape and talking about this experience (victim discarded measure is the sum of items: 1, 4, 18, 20, 23). The victim deservingness, victim credibility, and victim blame subscales differ in that the victim credibility scale assesses the level of belief that the situation actually occurred was rape, and the victim deservingness and victim blame subscales are based on the top-down assumption that rape took place. A high score on the victim credibility scale means a tendency to doubt whether the situation can be described as rape. The difference between the victim blame and victim deservingness scales is that a high victim blame score implies the assumption of the victim’s complicity for the rape situation, i.e., some behavior or attitude that resulted in the rape. On the other hand, a high score on the victim deservingness scale means that the respondent assumes that the victim deserved a given situation—it is not necessarily about active participation in initiating a given situation, but rather that the experience of rape should not be the basis for feeling hurt, because the victim will feel hurt, e.g., it “should have been” or it was not a big hurt [14]. The scales are enclosed as Appendix A.

Research conducted in various countries (including Australia and Singapore) also confirmed its significant intercultural usefulness [15]. Due to the fact that attitudes towards rape victims may have an impact on the quality of care provided to them in health care units, the ARVS scale is a valuable research tool applied in the field of victimology. The questionnaires were collected in direct contact with respondents. The return rate was 98.58% of the 1200 surveys we distributed. This was calculated according to generally accepted principles [16].

Ethical approval was granted by the Bioethics Commission of the University of Rzeszow (Resolution No. 13/02/2019). Participation was voluntary and anonymous (no personal identification was recorded), and the confidentiality of the participants was ensured. They were informed that the data obtained were anonymous and will be used only for research purposes. Details about the study objectives were provided in the questionnaire instruction and if the respondent returned the questionnaire, it indicated informed consent.

The quantitative variable analysis was performed by calculating the arithmetic mean ($\bar{x}$), standard deviation, median, and range. The qualitative variable analysis was performed by calculating the number and percentage of occurrences of each value. The comparison of the variables in two groups was performed using the *t*-test for independent samples and the values of the variables in three or more groups using the analysis of variance test (ANOVA). A synthetic description of the influence of the factors under consideration on individual ARVS measures (victim deservingness, victim credibility, victim blame, victim discarded, total score) was also made using the linear regression model. In the analysis a significance level of 0.05 was adopted. Therefore, all values of the statistical significance coefficient (*p*) below 0.05 were interpreted as showing significant relationships. The analysis was performed in Statistica v. 13 TIBCO Software Inc. (Palo Alto, CA, USA) (2017).

## 3. Results

To examine the factors that predict whether the respondents will have sympathetic attitudes towards rape victims and show understanding for them, we conducted a series of statistical analyzes (Table 2). Among others, demographic characteristics (gender, social environment, and religious commitment) were included. Table 2A shows the results of these analyzes, using total scores on the ARVS scale, as the dependent variable, and scores on subsequent subscales. The total score for the entire study population (1183 medical students) remains at the average level ($\bar{x}$ = 53.9), below the average score characterizing the scale, and higher than the lower quartile.

### 3.1. Univariate Analysis

According to assumptions on the ARVS scale, increases in scores are associated with less sympathetic attitudes towards rape victims. In this study, a relatively large variation in the results is related to the gender of the respondents (Table 2B), with male associated with higher scores on the ARVS, indicating that they held less sympathetic views of victims of rape than females (total score: 61.6 vs. 52.6). The higher scores obtained by men apply to both the entire ARVS scale and individual subscales, and in all cases the differences are statistically significant. Men showed a marked tendency to blame victims for the rape committed and to lower their credibility as witnesses. Table 2C also shows that the variance in attitudes towards rape victims is less explained by the attitude toward the religion and social environment of the respondents. The total scores on the ARVS scale obtained by persons who differ in attitudes towards religion remain at a similar level, and the differences between them did not reach the required level of statistical significance (*p* > 0.05). To some extent, this result is unexpected, as it would be expected that more religious people should show more empathy and understanding for all victims of sexual violence, including rape victims. Similarly, the social environment of the respondents turned out to be a variable that did not significantly differentiate their scores on the ARVS scale (Table 2D), both in relation to the total score and the scores of subsequent subscales (*p* > 0.05). This result can be treated as an indicator of the reduction of differences between the rural and urban social environment, which we have observed over the last few decades in Poland [17].

There are statistically significant differences between the results of the respondents’ depending on the field of study (Table 2E). The analysis of the average results summarized in Table 2E shows that the students of the medical faculty, i.e., future physicians, are in favor. They obtained the lowest results (total score: 49.7), which proves that they have shown much understanding and empathy for the victim’s rape. Respondents representing other study fields obtained very similar results, both in relation to the total score (midwifery and nursing: 54.1, other fields: 54.4) and individual subscales. The level of university study, that is, undergraduate versus graduate turned out to be an insignificant factor for the differentiation of the ARVS scores (Table 2F).

The results of the respondents who belong to both compared groups (total score: 54.2 vs. 53.5) did not differ statistically significantly (*p* > 0.05). However, an exception was found: the dimension of victim credibility was statistically significantly lower (17.3 vs. 18.0) for the graduate level of university study (*p* = 0.0059) (Table 2F).

The more positive attitude towards religion was, the lower empathy towards rape victims in terms of total score (very positive: 53.8, positive: 52.5, indifferent and negative: 50.5). For this relationship, statistical significance (*p* = 0.0157) was obtained not only for the total score, but also for the score for victim credibility (*p* = 0.0071), victim discarded (*p* = 0.0142) (Table 3A). This relationship was contrary to the preliminary expectations.

The social environment differentiated the attitude towards rape victims. Statistical significance was obtained for the total score and all subscales (*p* = 0.0003–0.0355). Rural women showed lower empathy for victims of rape, both in terms of the total score (53.5 vs. 51.1) and partial scores (Table 3B). In the case of women, differences related to the social environment were clearly revealed. Female students of the medical faculty were characterized by greater empathy; the results for the other two groups of faculties are very similar. Statistical significance was obtained for the total result as well as for all partial results (*p* = 0.0000–0.0165). Particularly alarming was the lower empathy towards rape victims in the case of future midwives and nurses due to their future professional commitment (total score: 53.8 vs. 47.1) (Table 3C).

The level of the university study differentiated attitudes towards rape victims in two areas: victim deservingness (*p* = 0.0283) and victim credibility (*p* = 0.0439). The impact can be described as the opposite. A higher dimension of victim deservingness was found in female graduate students (9.9 vs. 9.4), and a higher dimension of victim credibility in female undergraduate students (17.5 vs. 17.0) (Table 3D).

In general, no statistically significant correlations were found between the differentiating factors considered and the dimension of attitudes toward rape victims among men (Table 4A,B,D). The reason for this may be the smaller size of the compared groups. However, there is one exception (Table 4C). Male students of medicine were characterized by greater empathy, achieved a lower total score than students of nursing, obstetrics/other study fields of (55.9 vs. 62.7/63.0, *p* = 0.0069), as well as lower partial results in terms of victim deservingness (9.8 vs. 20.7/20.4, *p* = 0.0012) and victim credibility (18.5 vs. 20.7/20.4, *p* = 0.0362). Female medical students (as presented above) were also characterized by higher empathy.

### 3.2. Linear Regression Model

For each of the ARVS dimensions, both partial and total scores, a linear regression model was constructed containing only statistically significant differentiating factors. The following dichotomous characteristics were taken into account as the initial set of independent factors: gender (male vs. female), attitude towards religion (very positive and positive vs. neutral and negative), social environment (rural vs. urban), field of study (medicine vs. other), and the level of education (graduate vs. undergraduate). This division of differentiating factors was selected on the basis of the results of the univariate analyzes presented above and the presence of statistically significant relationships. Furthermore, second-degree interactions between gender (a very strong differentiating factor) and other factors were introduced into the model. Ultimately, five independent factors and four interactions between them were introduced as potential factors differentiating the ARVS dimension. Using the progressive stepwise regression procedure, optimal models were found containing only statistically significant factors.

The presence of common characteristics for the differentiating factors was found (Table 5):Up to 10.1% (R^2^ ≤ 10.1%) of the variability in the dimension of attitudes towards rape victims was explained by the variables introduced into the model (victim deservingness—5.8%, victim credibility—7.6%, victim discarded—3.2%, victim blame—10.1%, and total score—9.5%).The dimension of ARVS: all partial results and the total result are statistically significantly influenced by gender and the field of study, while in terms of gender (men vs. women), women are characterized by lower results assessed in the ARVS scale, i.e., higher empathy in relation to victims of rape (coefficient ß is positive in all cases), and in the case of the field of study (medicine vs. other), lower ARVS scores, i.e., higher empathy distinguishes respondents studying medicine (coefficient ß is negative in all cases).

Based on the value of the standardized regression coefficient ß, it can be concluded that gender has a greater impact on the ARVS dimension than the field of study (in all cases the absolute value of the coefficient ß is closer to zero in the case of the field of study)Additionally, on the basis of the regression models constructed for the partial ARVS scores and the total ARVS score, the importance of other differentiating factors or the interaction of these factors with gender was demonstrated.

In the model describing the influence of differentiating factors on the dimension of victim deservingness (*p* = 0.0000), the following independent factors had a statistically significant influence: gender (*p* = 0.0000), study field (*p* = 0.0001) and interaction between gender and university study level (*p* = 0.0136) (Table 5A). The interaction means that in women and men, the impact of the study level on the dimension of victim deservingness was opposite among men, students from the graduate study level had a lower dimension of victim deservingness (less empathy), and the opposite was observed among women (Figure 1A).

In addition to the previously described influence of gender (*p* = 0.0000) and the study field (*p* = 0.0002) on the dimension of victim credibility (women and medical students were characterized by a higher level of empathy toward rape victims in this respect), a significant influence of the social environment was revealed, but in interaction with gender (*p* = 0.0034). For the entire model that describes the influence of differentiating factors, statistical significance was found (*p* = 0.0000). Among men, who came from a rural social environment, decreased dimension of victim credibility (i.e., an increased level of empathy) was found, and the opposite was true for women (Table 2B and Figure 1B). Female students from rural areas were characterized by a higher dimension of victim credibility, i.e., a lower level of empathy for rape victims (Figure 1B).

In the model, for a victim blame dimension (*p* = 0.0000), two factors that differentiate this dimension were indicated: gender and study field (women and medical students were characterized by a higher level of empathy toward rape victims in this respect) (Table 5C).

In the model describing the influence of factors differentiating the dimension of victim discarded (*p* = 0.0000), apart from gender (*p* = 0.0000) and the study field (*p* = 0.0013) (women and medical students were characterized by a higher level of empathy toward rape victims in this respect), the social environment was also indicated as an important one (*p* = 0.0262, students from a rural social environment had a significantly higher dimension of victim-discarded, which means a lower level of empathy). Furthermore, there was also a statistically significant interaction of gender with attitude towards religion (*p* = 0.0302)—among men with a positive/very positive attitude towards religion, greater empathy was found (lower dimension of victim discarded), and among women with a positive/very positive attitude towards religion, the lower level of empathy (i.e., the higher dimension of victim discarded) (Table 5D and Figure 1C).

In addition to the previously described influence of gender (*p* = 0.0000) and study field (*p* = 0.0000) on the total score dimension (women and medical students were characterized by a higher level of empathy toward rape victims in this respect), a significant influence of the social environment was visible but in interaction with gender (*p* = 0.0034). For the entire model that describes the influence of differentiating factors, the statistical significance was achieved (*p* = 0.0000). Among men, both of rural or urban background, the diversity of the victim credibility dimension was small, and women of rural background were characterized by a high dimension of victim credibility (lower level of empathy) compared to those of urban background (Table 5E and Figure 1D).

## 4. Discussion

Knowledge about the role that some basic socio demographic characteristics play is very important because it can be used in the design of future prevention and intervention programs. Thus far, no such research has been conducted in Poland among medical students. In our study, in total, mean scores on the ARVS scale indicate that a significant number of all respondents have moderately positive attitudes towards rape victims, although female students had a greater accepting attitude towards survivors of rape than males. This finding is consistent with studies conducted in groups of medical students in different countries [18,19,20]. Although there are exceptions, a study of Indian medical students [21] or Turkish students [16] has not shown gender to differentiate ARVS scores. The gender of prospective physicians or other medical specialists may have been an influencing factor in their interactions with rape victims seeking professional help. Therefore, some authors argue for the inclusion of rape education to the medical student curriculum in the hope of challenging misconceptions by providing factual information and improving future rape victim management in medical and health disciplines [19]. However, the influence of gender on attitudes of this kind should not be separated from the cultural conditions in which they function. For example, Asian college students may be more likely to view rape victims in a negative context than their Caucasian peers due to Asian cultural traditions which endorse a patriarchal structure in which the status of women is low. Indirectly, this suggests that Asian women may be more likely to underreport sexual assaults due to a possible failure to recognize rape as a sexual attack and due to fear of negative repercussions or self-blame [22]. ‘Asian’ is a very broad term, highly diverse, grouping of ethnicities, and the cultural traditions are extraordinarily different. Similar results are obtained in studies on acceptance of rape myths. For example, Australian male university students still have a significant tendency, greater than their female counterparts, to accept rape myths and to diminish the seriousness of rape. However, the results also imply that the determination of rape attributions is a factor of both the level of acceptance of the rape myth and the gender [23]. The authors of this study believe that the difference in attitudes toward rape victims may also be due to the fact that women are more susceptible to this type of crime than men. Ninety percent of rape victims are women [24]. Anderson and Quinn found that, male victims were viewed more negatively than female victims [21]. In addition, the results of other studies highlight the importance of examining the interactive effects of demographic variables when analyzing complex relationships that predict attitudes towards rape victims [25,26]. In our own research, interactions were found between gender (a very strong differentiating factor) and factors such as social environment, field of study, and attitude toward religion.

Contrary to the authors’ expectations, the attitude toward religion of the respondents turned out to be a variable which did not differentiate attitudes towards rape victims in the entire study group and especially in the case of men. In terms of religion, they form a relatively homogeneous group (i.e., Roman Catholic) that follows a similar hierarchy of moral values. The literature shows that religious people tend to be more judgmental and less empathetic, especially as the degree of fundamentalism increases. In the literature on the subject, there is a little research on the relationship between attitudes toward rape victims and the depth of religious commitment or religious beliefs. However, in one of the few studies by Sheldon and Parent (2002) it was found that those who scored the highest on measures of sexism and religious fundamentalism had the highest levels of rape myth acceptance and victim-blame attitudes, and the majority of study participants more or less blamed the victim [27]. In a more recent study it was found that a high degree of authoritarianism was more closely related to the acceptance of rape myths and a negative outlook toward rape victims than Christian fundamentalism [28]. However, among the female medical students surveyed, a more positive attitude of women towards religion was associated with lower empathy towards rape victims in terms of the total score of the ARVS, victim credibility, and victim discarded subscales. This relationship was contrary to the preliminary expectations. The interaction between attitude toward religion and gender is opposite in terms of victim discarded dimension, both in women and men.

Our study shows that medical students living in an urban social environment achieved slightly lower results than their rural colleagues (53.2 vs. 54.3), which shows a more positive attitude toward rape victims, but this difference did not turn out to be statistically significant (*p* > 0.05). Furthermore, other studies did not report any significant relationship of attitudes toward rape victims with residential status [20,21]. This finding may indicate the leveling of sociocultural differences between the urban and rural environment in recent decades. To some extent, this is confirmed by the fact that earlier studies have confirmed that students from rural areas used to be more conservative with respect to attitudes of accountability for rape than students from urban areas [29,30]. It is interesting to note that attitudes towards rape are not significantly related to residents of urban or rural areas, but the volume of rape myths they carry may differ, as evidenced by other study [31]. This may be because, in general, people belonging to rural backgrounds are more inclined to tolerate sexual violence and carry more rape myths. Additionally, gender moderated the relations between cold-heartedness (a trait of psychopathic personality) and acceptance of rape myths, such effects were significantly stronger for women [32]. An example of a myth may be the belief that “a scantily dressed woman provokes rape” [32]. The question based on this myth is included in the part of the ARVS scale that evaluates the level of belief that the situation, in fact, was a victim credibility. In the authors’ own research, based on the linear regression model, it was shown that female medical students from rural areas have a higher dimension of victim credibility, that is, they have a weaker conviction that the specific situation was rape. The role of social environment is different in terms of victim credibility and total score in women and men.

The level of university study differentiated the dimension of victim credibility in the entire study group. Students at the higher levels of study achieved a statistically significantly lower range of victim credibility (higher level of empathy). This relationship was obtained for the entire study group and especially for women. On the other hand, the victim deservingness dimension was statistically significantly lower in women from lower levels of study. Additionally, there is an interaction between gender and the level of university education—in women and men, the impact of the level of study on the victim deservingness dimension was opposite—among men, students from the graduate level of education presented a lower dimension of victim deservingness, and vice versa among women. In the literature there are examples of studies showing that undergraduates, especially men, more often exhibit negative attitudes than graduates [18]. The field of the study brings a slightly greater diversity in this respect. In addition to gender, the field of university study had a statistically significant influence on the ARVS results, both total and partial. It was the second most important differentiating factor after gender.

There are no cut-off points to distinguish a descriptive assessment of attitudes towards rape victims as positive or negative. It was found that if the average value of the total score for the entire studied group is 53.9, it is below the average value characterizing the scale, and above the value corresponding to the lower quartile, and it cannot be unequivocally determined that the attitude is positive; therefore, it was assessed as moderately positive. The students of medicine (came physicians) presented the most favorable attitudes toward rape victims. In the literature, examples of research can be found, the results of which confirm this finding. For example, studies among Greek students showed that law and applied sciences students seem to have accepted more conservative attitudes on the definition of the concept of rape, while humanities and social sciences students may provide a fertile ground for less conservative ideas towards rape and a more sensitive approach to rape incidents, without stereotypical elements and rape myths [26]. There are examples of other, not yet discussed, differentiating factors. There has been shown to be a significant positive relationship between modern racism and blame for rape victims, while modern racism had a significant negative relationship with perpetrator blame and rape [33].

The role of gender in differentiating attitudes towards rape victims is very complex and ambiguous [34]. In this study, gender was the most important differentiating factor. Its direct impact on the attitude towards rape victims declared by medical students was observed, as well as on interacting with other differentiating factors, such as level of study, social environment, attitude toward religion, which determined the dimension of attitudes toward rape victims. The field of medical studies was the second most important differentiation factor. However, it has not been observed to interact significantly with other differentiating factors.

## 5. Conclusions

There were two main factors differentiating attitudes towards rape victims: gender and the field of medical studies. Medical students surveyed had moderately positive attitudes towards rape victims. Among them, men have consistently somewhat negative attitudes toward rape victims than women who make more pro-victim judgments. The analysis of the results also shows that the variance in attitudes toward rape victims is less explained by the religious commitment and social environment of the respondents. Although the level of studies (undergraduate vs. graduate) turned out to be a variable that slightly differentiated attitudes toward rape victims, the field of study is of significant importance. Future doctors showed more empathy and understanding for rape victims than future nurses and midwives.

### 5.1. Defining the Direction of Future Research

Finally, it should be emphasized that despite the growing public awareness of the problem of rape, this study shows that even among medically educated people there are attitudes characterized by prejudices and a tendency to blame the victim. Although this type of tendency is moderate in terms of intensity, it exists regardless of gender, field of study, social environment, etc. One of the practical implications of this research is the proposal to prepare and implement educational programs for medical students oriented toward changing their attitudes, which would contribute to reducing the stigma surrounding rape victims.

Future research should be conducted to explain why medical students have a higher level of empathy for rape victims. The role of factors such as the higher level of requirements for entrance examinations, the content of the curricula, the teaching model during studies, and personal predisposition to the medical profession should also be clarified.

Future research should consider other factors differentiating the dimension of attitudes towards rape victims, such as: socio-cultural differences between the urban and rural environment, the degree of authoritarianism, and racism. The use of the linear regression model makes it possible to analyze the influence of gender or religiosity with the differentiating factors mentioned above among the respondents. Additionally, it is advisable to take into account whether the victim status: refugee, different nationality, war victim differentiate the dimension of attitudes towards rape victims.

### 5.2. Limitations

Although the current study provided many interesting and notable findings, it is not without its limitations, which are typical for many cross-sectional studies. Primarily, this type of study design shows exposure and outcome at the same point in time, so that we cannot formulate a cause and effect relationship. Additionally, it should be noted that the findings the student samples may not be applicable to other populations. Furthermore, within this study, the relatively small sample size of male students resulted from the large majority of women among medical students. The study is subject to possible response bias because the subjects may feel that they have to respond in a socially acceptable manner. It is understandable that the authors have no way to verify whether these self-reported attitudes are consistent with the behavior of the respondents. Any study based on self-reported information is subjected to reporting errors, missed values, and biases. Since this study touches on sensitive issues, the possibility of underestimation cannot be excluded. Despite these limitations, the authors hope that the study provides valuable insight on knowledge and attitudes toward rape victims presented by medical students.

## Figures and Tables

**Figure 1 ijerph-19-05896-f001:**
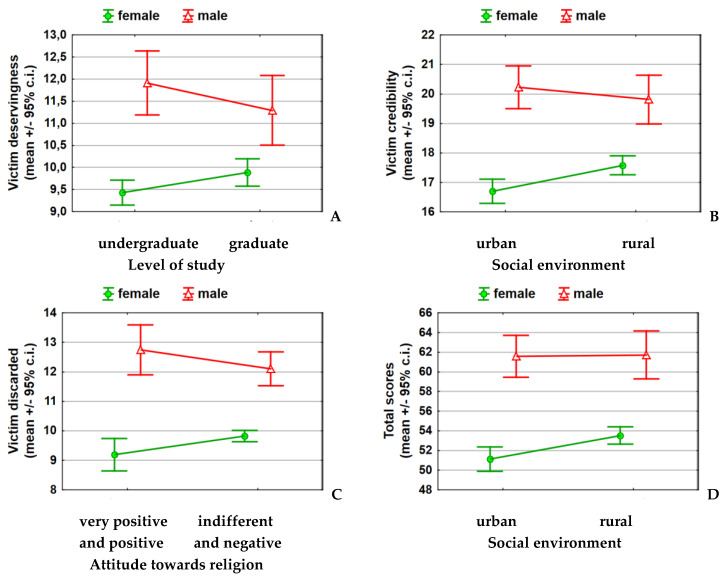
Linear regression model containing only statistically significant factors differentiating the ARVS dimension illustration of second−degree interactions. (**A**) Victim deservingness: gender and level of study; (**B**) Victim credibility: gender and social environment; (**C**) Victim discarded: gender and attitude toward religion; (**D**) Total stores: gender and social environment.

**Table 1 ijerph-19-05896-t001:** Participants’ demographics.

Variable	*N*			%		
Gender						
Female	1016			85.9		
Male	167			14.1		
Variable	Female *N*	Female %	Male *N*	Male %	Total *N*	Total *N*%
Level of study						
Undergraduate	518	43.8	102	8.6	620	52.4
Graduate	498	42.1	65	5.5	563	47.6
Field of study						
Midwifery	148	12.5	0	0	148	12.5
Nursing	227	19.2	11	0.9	238	20.1
Physiotherapy	390	33.0	97	8.2	487	41.2
Dietetics	140	11.9	11	0.9	151	12.8
Medicine	75	6.3	32	2.7	107	9.0
Public health	33	2.8	8	0.7	41	3.5
Emergency medical services	3	0.2	8	0.7	11	0.9
Social environment						
Urban	391	33.1	99	8.3	490	41.4
Rural	625	52.8	68	5.8	693	58.6
Attitude toward religion						
Very positive	266	22.5	22	1.8	288	24.3
Positive	617	52.2	94	7.9	711	60.1
Indifferent	111	9.4	42	3.5	153	12.9
Negative	22	1.7	9	0.9	31	2.6

*N*—number, %—percent.

**Table 2 ijerph-19-05896-t002:** ARVS scores in the entire study group.

A. ARVS Scores (*N* = 1183)	Mean (95% c.i.)	SD	Median	Range
Victim deservingness	9.9 (9.7–10.1)	3.5	9	6–28
Victim credibility	17.6 (17.4–17.9)	4.1	18	7–30
Victim blame	16.2 (15.9–16.5)	4.5	16	7–31
Victim discarded	10.1 (9.9–10.3)	3.1	10	5–22
Total score	53.9 (53.2–54.6)	12.0	54	25–105
ARVS scores	B. Gender (*t*-test for independent samples)			*p*
	Female (*N* = 1016)	Male (*N* = 167)		
	Mean (95% c.i.)			
Victim deservingness	9.7 (9.4–9.9)	11.7 (11.1–12.2)		0.0000
Victim credibility	17.2 (17.0–17.5)	20.1 (19.5–20.6)		0.0000
Victim blame	16.0 (15.7–16.3)	17.6 (17.1–18.2)		0.0000
Victim discarded	9.7 (9.6–9.9)	12.3 (11.8–12.8)		0.0000
Total score	52.6 (51.9–53.3)	61.6 (60.0–63.2)		0.0000
ARVS scores	C. Attitude towards religion (analysis of variance test)			*p*
	Very positive (*N* = 288)	Positive (*N* = 711)	Indifferent and negative (*N* = 184)	
	Mean (95% c.i.)			
Victim deservingness	9.9 (9.5–10.4)	9.8 (9.6–10.1)	10.3 (9.7–10.8)	0.3157
Victim credibility	17.9 (17.4–18.4)	17.6 (17.3–17.9)	17.4 (16.7–18.1)	0.3590
Victim blame	16.3 (15.8–16.8)	16.2 (15.9–16.5)	16.0 (15.3–16.7)	0.7533
Victim discarded	10.1 (9.8–10.5)	10.1 (9.8–10.3)	10.2 (9.7–10.7)	0.9169
Total score	54.3 (53.0–55.6)	53.7 (52.9–54.6)	53.8 (51.8–55.9)	0.7882
ARVS scores	D. Social environment (*t*-test for independent samples)			*p*
	Urban (*N* = 490)	Rural (*N* = 693)		
	Mean (95% c.i.)			
Victim deservingness	9.9 (9.6–10.2)	10.0 (9.7–10.2)		0.7369
Victim credibility	17.4 (17.0–17.8)	17.8 (17.5–18.1)		0.1131
Victim blame	16.0 (15.6–16.4)	16.4 (16.0–16.7)		0.1685
Victim discarded	10.0 (9.7–10.2)	10.2 (10.0–10.4)		0.1757
Total score	53.2 (52.1–54.4)	54.3 (53.5–55.2)		0.1286
ARVS scores	E. Field of study (analysis of variance test)			*p*
	Midwifery and nursing (*N* = 386)	Medicine (*N* = 107)	Other fields (*N* = 690)	
	Mean (95% c.i.)			
Victim deservingness	10.1 (9.8–10.5)	9.1 (8.5–9.6)	10.0 (9.7–10.2)	0.0187
Victim credibility	17.5 (17.1–17.9)	16.6 (15.7–17.5)	17.9 (17.5–18.2)	0.0100
Victim blame	16.5 (16.1–17.0)	14.7 (13.9–15.4)	16.2 (15.9–16.6)	0.0006
Victim discarded	9.9 (9.6–10.2)	9.5 (8.8–10.1)	10.3 (10.1–10.6)	0.0056
Total score	54.1 (52.9–55.2)	49.7 (47.3–52.2)	54.4 (53.5–55.3)	0.0008
ARVS measures	F. Level of study (*t*-test for independent samples)			*p*
	Undergraduate (*N* = 620)	Graduate (*N* = 563)		
	Mean (95% c.i.)			
Victim deservingness	9.8 (9.6–10.1)	10.0 (9.8–10.3)		0.3084
Victim credibility	18.0 (17.6–18.3)	17.3 (16.9–17.6)		0.0059
Victim blame	16.2 (15.9–16.6)	16.1 (15.8–16.5)		0.7258
Victim discarded	10.2 (10.0–10.5)	10.0 (9.7–10.2)		0.1980
Total score	54.2 (53.3–55.2)	53.5 (52.4–54.5)		0.2610

c.i.—confidence interval, *N*—number, *p*—statistical significance coefficient, SD—standard deviation.

**Table 3 ijerph-19-05896-t003:** ARVS scores among the surveyed women.

ARVS scores	A. Attitude towards religion (analysis of variance test)			*p*
	Very positive (*N* = 266)	Positive (*N* = 617)	Indifferent and negative (*N* = 133)	
	Mean (95% c.i.)			
Victim deservingness	9.8 (9.4–10.2)	9.6 (9.3–9.8)	9.6 (9.0–10.3)	0.6298
Victim credibility	17.7 (17.2–18.2)	17.2 (16.9–17.5)	16.4 (15.5–17.2)	0.0071
Victim blame	16.3 (15.7–16.8)	16.0 (15.6–16.3)	15.3 (14.5–16.2)	0.1057
Victim discarded	10.0 (9.7–10.3)	9.7 (9.5–10.0)	9.2 (8.6–9.7)	0.0142
Total score	53.8 (52.5–55.2)	52.5 (51.6–53.4)	50.5 (48.1–53.0)	0.0157
ARVS scores	B. Social environment (*t*-test for independent samples)			*p*
	Urban (*N* = 391)	Rural (*N* = 625)		
	Mean (95% c.i.)			
Victim deservingness	9.5 (9.1–9.8)	9.8 (9.5–10.0)		0.0221
Victim credibility	16.7 (16.3–17.1)	17.6 (17.3–17.9)		0.0004
Victim blame	15.6 (15.1–16.1)	16.2 (15.8–16.5)		0.0355
Victim discarded	9.4 (9.1–9.7)	10.0 (9.7–10.2)		0.0003
Total score	51.1 (49.9–52.4)	53.5 (52.6–54.4)		0.0007
ARVS scores	C. Field of study (analysis of variance test)			*p*
	Midwifery and nursing (*N* = 375)	Medicine (*N* = 75)	Other fields (*N* = 566)	
	Mean (95% c.i.)			
Victim deservingness	10.1 (9.7–10.4)	8.7 (8.0–9.4)	9.5 (9.2–9.8)	0.0004
Victim credibility	17.4 (17.0–17.8)	15.8 (14.6–16.9)	17.3 (17.0–17.6)	0.0165
Victim blame	16.5 (16.0–17.0)	13.9 (12.9–14.8)	15.9 (15.5–16.3)	0.0000
Victim discarded	9.8 (9.5–10.1)	8.8 (8.0–9.5)	9.8 (9.6–10.1)	0.0053
Total score	53.8 (52.7–55.0)	47.1 (44.2–50.1)	52.5 (51.6–53.5)	0.0000
ARVS measures	D. Level of study (*t*-test for independent samples)			*p*
	Undergraduate (*N* = 518)	Graduate (*N* = 498)		
	Mean (95% c.i.)			
Victim deservingness	9.4 (9.2–9.7)	9.9 (9.6–10.2)		0.0283
Victim credibility	17.5 (17.1–17.8)	17.0 (16.6–17.4)		0.0439
Victim blame	16.0 (15.6–16.4)	16.0 (15.5–16.4)		0.8507
Victim discarded	9.8 (9.5–10.0)	9.7 (9.4–10.0)		0.6402
Total score	52.6 (51.7–53.6)	52.6 (51.5–53.6)		0.7045

c.i.—confidence interval, *N*—number, *p*—statistical significance coefficient.

**Table 4 ijerph-19-05896-t004:** ARVS scores among the surveyed men.

ARVS scores	A. Attitude towards religion (analysis of variance test)			*p*
	Very positive (*N* = 22)	Positive (*N* = 94)	Indifferent and negative (*N* = 51)	
	Mean (95% c.i.)			
Victim deservingness	11.6 (10.1–13.1)	11.5 (10.8–12.3)	11.9 (11.0–12.9)	0.5922
Victim credibility	20.2 (18.8–21.5)	20.0 (19.3–20.7)	20.1 (19.0–21.1)	0.9325
Victim blame	16.5 (14.9–18.2)	17.8 (17.0–18.5)	17.7 (16.7–18.7)	0.1974
Victim discarded	11.6 (10.1–13.2)	12.2 (11.6–12.8)	12.7 (11.9–13.6)	0.3460
Total score	60.0 (55.7–64.2)	61.6 (59.4–63.7)	62.5 (59.5–65.5)	0.6179
ARVS scores	B. Social environment (*t*-test for independent samples)			*p*
	Urban (*N* = 99)	Rural (*N* = 68)		
	Mean (95% c.i.)			
Victim deservingness	11.6 (10.9–12.4)	11.7 (10.9–12.5)		0.7424
Victim credibility	20.2 (19.5–20.9)	19.8 (19.0–20.6)		0.4340
Victim blame	17.4 (16.7–18.2)	17.8 (17.0–18.7)		0.3587
Victim discarded	12.3 (11.6–12.9)	12.4 (11.6–13.1)		0.8789
Total score	61.6 (59.4–63.7)	61.7 (59.3–64.2)		0.8386
ARVS scores	C. Field of study (analysis of variance test)			*p*
	Midwifery and nursing (*N* = 11)	Medicine (*N* = 32)	Other fields (*N* = 124)	
	Mean (95% c.i.)			
Victim deservingness	11.3 (8.7–13.8)	9.8 (8.8–10.8)	12.2 (11.6–12.8)	0.0012
Victim credibility	20.7 (18.8–22.6)	18.5 (17.0–19.9)	20.4 (19.8–21.0)	0.0362
Victim blame	18.7 (16.8–20.7)	16.5 (15.2–17.9)	17.8 (17.1–18.4)	0.1526
Victim discarded	12.0 (10.1–13.9)	11.1 (9.9–12.2)	12.6 (12.1–13.2)	0.0638
Total score	62.7 (56.8–68.7)	55.9 (51.9–59.9)	63.0 (61.2–64.8)	0.0069
ARVS measures	D. Level of study (*t*-test for independent samples)			*p*
	Undergraduate (*N* = 102)	Graduate (*N* = 65)		
	Mean (95% c.i.)			
Victim deservingness	11.9 (11.2–12.6)	11.3 (10.5–12.1)		0.3221
Victim credibility	20.4 (19.7–21.1)	19.5 (18.6–20.4)		0.0893
Victim blame	17.6 (16.9–18.3)	17.6 (16.6–18.5)		0.8200
Victim discarded	12.4 (11.8–13.0)	12.1 (11.3–12.9)		0.5535
Total score	62.4 (60.4–64.4)	60.4 (57.7–63.1)		0.2910

c.i.—confidence interval, *N*—number, *p*—statistical significance coefficient.

**Table 5 ijerph-19-05896-t005:** Linear regression model containing only statistically significant factors differentiating the ARVS dimension.

Independent features	A. Victim deservingness *R*^2^ = 5.8% *F* = 24.2 *p* = 0.0000		
	*B* (95% c.i.)	*p*	*β*
Gender (male vs. female)	2.119 (1.559; 2.679)	0.0000	0.21
Field of study (medicine vs. other)	−1.361 (−2.039; −0.684)	0.0001	−0.11
Gender (male vs. female) × Level of study (graduate vs. undergraduate)	−0.486 (−0.873; −0.100)	0.0136	−0.07
Independent features	B. Victim credibility *R*^2^ = 7.6% *F* = 32.3 *p* = 0.0000		
	*B* (95% c.i.)	*p*	*β*
Gender (male vs. female)	3.011 (2.350; 3.671)	0.0000	0.25
Field of study (medicine vs. other)	−1.538 (−2.346; −0.730)	0.0002	−0.11
Gender × Social environment (rural vs. urban)	−0.704 (−1.175; −0.233)	0.0034	−0.08
Independent features	C. Victim blame *R*^2^ = 3.2% *F* = 19.4 *p* = 0.0000		
	*B* (95% c.i.)	*p*	*β*
Gender (male vs. female)	1.877 (1.137; 2.616)	0.0000	0.14
Field of study (medicine vs. other)	−2.023 (−2.921; −1.125)	0.0000	−0.13
Independent features	D. Victim discarded *R*^2^ = 10.1% *F* = 32.9 *p* = 0.0000		
	*B* (95% c.i.)	*p*	*β*
Gender (male vs. female)	3.057 (2.499; 3.616)	0.0000	0.34
Social environment (rural vs. urban)	0.401 (0.048; 0.754)	0.0262	0.06
Field of study (medicine vs. other)	−0.997 (−1.604; −0.389)	0.0013	−0.09
Gender (male vs. female) × Attitude toward religion (very positive and positive vs. indifferent and negative)	−0.527 (−1.004; −0.051)	0.0302	−0.07
Independent features	E. Total scores *R*^2^ = 9.5% *F* = 41.1 *p* = 0.0000		
	*B* (95% c.i.)	*p*	*β*
Gender (male vs. female)	9.759 (7.868; 11.649)	0.0000	0.28
Field of study (medicine vs. other)	−5.897 (−8.209; −3.584)	0.0000	−0.14
Gender (male vs. female) × Social environment (rural vs. urban)	−1.591 (−2.939; −0.244)	0.0206	−0.06

Linear regression model. *R*^2^—coefficient of determination, *F*—test statistic and *p* value for significance of whole model, *B*—regression coefficient with 95% c.i., *p*—value for significance of each regression coefficient, *β*—standardize regression coefficient, ×—interaction between factors.

## Data Availability

Data are available from the authors.

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
