# Peer review of "Selected Correlates of Attitudes towards Rape Victims among Polish Medical Students"

_ijerph, 2022, doi:10.3390/ijerph19105896_

Round 1

Reviewer 1 Report

The phrase R257-261 must be taken to R61 for the aim and objectives.

p must be written in italics throughout the text.

The text must be written in palatine linotype font 10 from R114-146 and from R148-151.

Figure 1E is missing.

Please replace social milieu with social environment.

Please rephrase the conclusions to be more relevant to the results of the study.

References 6, 11 and 30 must be corrected.

Author Response

Response to the Reviewer 1 comments:

Our responses to comments and suggestions made by the reviewer are marked in red.

  1. The phrase R257-261 must be taken to R61 for the aim and objectives.

We moved the indicated fragment (‘The purpose of this study was to assess and understand the variation of the attitudes of medical students towards victims of rape. In particular, the authors wanted to determine relationship between some basic sociodemographic variables and these attitudes’) from position 257 to position 61.

  1. pmust be written in italics throughout the text.

 p was changed into p.

  1. The text must be written in palatine linotype font 10 from R114-146 and from R148-151.

The font has been corrected according to the instructions.

  1. Figure 1E is missing

The incorrect reference Figure 1E has been replaced with the correct reference Figure 1D.

  1. Please replace social milieu with social environment.

The term 'social milieu' was replaced with the term 'social environment'.

  1. Please rephrase the conclusions to be more relevant to the results of the study.

‘There were two main factors differentiating attitudes towards rape victims: gender and the field of medical studies. The surveyed medical students presented moderately positive attitudes towards rape victims. Among them, men have consistently somewhat negative attitudes toward rape victims than women who make more pro-victim judgments. The analysis of results also shows that the variance of attitudes towards rape victims is less explained by religious commitment and social environment of the respondents. Although the level of studies (undergraduate vs. graduate) turned out to be a variable that slightly differentiated attitudes towards rape victims, the field of study is of significant importance. The future doctors showed more empathy and understanding for rape victims than the future nurses and midwives’.

  1. References 6, 11 and 30 must be corrected.

References were corrected, and appropriate punctuation marks were used.

On behalf of all the authors, I would like to thank the Reviewer for his/her suggestions and proposed changes. Their implementation will certainly contribute to increasing the scientific value of the reviewed article.

Reviewer 2 Report

This is a very interesting paper and one that was done at an opportune time, given the location of the study and the probability that the medical students in this sample are likely to be central to the treatment of a very large number of rape and war victims over the next few years.  The English is fine but a little stilted in places – a quick re-read might be advised to ensure clarity.

Line 66: what proportion of the enrolments at each point were male? (this information would reinforce the point in line 73 … if females are more likely to participate)

The gender imbalance of both the sample and the student population in the chosen faculties at University of Rzeszow has been well identified.  This is important because this might be expected to result in some level of inter-gender exchange of attitudes depending on the gender balance of the faculty teaching staff (because modifications of view may be influenced by views implicit in their teaching).  The use of “advantage” in line 74 does not feel quite right … perhaps intended is the gender balance of enrolments?

Table 1: this would be greatly helped if the demographics were split by sex.  This can be derived of course from the other tables but having it included here would be useful because features like the predominance of females in nursing/midwifery, greater rurality of females, and relative larger proportion of males in medicine would be clear. It would also be clear that 61% of the males and 51% of the females were undergraduates.  The average age and range is interesting – perhaps add average for males and for females. Also suggest putting urban above rural since this is the order in the other tables.

Section 2.2 is well described.  It would help to have a copy of the questionnaire as an appendix (both original and a translation if possible).  It would help to also have a small table of definitions because the labels on the measures are a little ambiguous as they stand (e.g. ‘deservingness’ and ‘discarded’ can be misread). A clear description of what they mean and also perhaps how they relate to the categories that are used in other literature would help (especially since Ward’s AVRS is used as the basis).

Lines 132-133: agree that it is reasonable to expect this to be unexpected on ethical grounds but in practice this is unlikely – generally I think the literature repeatedly shows that religious people tend to be more judgmental and less empathetic, especially as the degree of fundamentalism increases. The work of Sheldon and Parent and of Carr is cited (lines 290-300). An interaction between rurality and religiosity might be expected.  It may well be that the driver is the degree of authoritarianism rather than the ethics, since this would explain both lack of empathy among the religious and among the secular authoritarian (suggested by Carr – and an important dimension) – but I do realise this was not a dimension collected – may be a topic for further research. Certainly, the findings (lines 152-156 for example) show this is clear with respect to religion.  It does look as though this is a topic worth further work because of the finding by gender (lines 244-246) is of considerable interest.

One thought that does occur (lines 134-138) is whether the difference between urban and rural might be at least in part explained by differences in social class? To what extent is the reduction (line 136-7) in differentiation of rural and urban social environments the result of growing gentrification of the areas relatively close top the city?  It might be good to cite some research in support (not suggesting it is wrong – quite the opposite – but interesting) – one feature of the urban/rural dichotomy is that there has tended to be an embedding of historical attitudes intergenerationally but which is modified by population mobility (the students have moved to an urban environment but they will still carry with them a level of historical views that may dilute as they move through the years of study).

Table 2, 3, 4 are very useful. Just a tiny point: there seems to be a formatting slip in tables 3 and 4 (first category is bolded and underscored for columns 1-3)

Lines 169-173 are still referring to table 3 – should this precede table 4 to keep the women together?  This will, of course, make the analysis of men stand out as a bit thin, but the para explains that well.

Excellent discussion section. Be carefully of stereotyping (lines 273-275) – line 272-3 makes a vitally important point.  Just that ‘Asian’ is a very broad, highly diverse, grouping of ethnicities and the cultural traditions are extraordinarily diverse – although the point about under-reporting due to social consequences is well made for many of the cultures in question.

Lines 281-4: again, a very pertinent point. Does this study investigate differing attitudes depending on sex of the victim?  The way the text flows from line 281 implies that this information was collected – if not then perhaps prefix the sentence in line 284 with “Anderson and Quinn found that … [17]”

Conclusion: thank you for a conclusion that is actually a conclusion – very good suggestions for further study, very good recognition of limitations.  If the opportunity arises for further study it would be interesting to add in attitudes to rape when the victim is a refugee or is foreign born.  This may show deep seated racist or religious attitudes which could mean not all victims would be treated equally (just as a topical example – would attitudes vary if the victim was Syrian or Ukrainian? – would the characteristics of the perpetrator influence the attitudes towards the victim? … touched on in this paper quite correctly – or if the person was a victim of the clergy [a thought provoked by the differences in attitudes by religiosity]).

Author Response

Response to the Reviewer 2 comments:

Our responses to comments and suggestions made by the Reviewer and changes we implemented in the text are marked in red.

The English is fine but a little stilted in places – a quick re-read might be advised to ensure clarity’

The text was read and corrected by a native speaker in order to make English more fluent.

Line 66: what proportion of the enrolments at each point were male? (this information would reinforce the point in line 73 … if females are more likely to participate)

The indicated data has been supplemented. At the University of Rzeszow, the number of all students in medical faculties as of December 31, 2019 was 2,785 (women constituted 79.9% – 2,227 persons), and as of December 31, 2020 - 2,907 (women constituted 79.1% – 2,301 persons).

The use of “advantage” in line 74 does not feel quite right … perhaps intended is the gender balance of enrolments?

There is no gender criterion when qualifying for studies. The sentence was redrafted as follows: ‘ The disproportion between the genders reflects the great advantage of women in medical studies at the University of Rzeszow, turned into: In medical studies at the University of Rzeszow occurs the disproportion between the genders’.

Table 1: this would be greatly helped if the demographics were split by sex. 

Table 1 has been redrafted and the indicated data have been entered.

The average age and range is interesting – perhaps add average for males and for females. Also suggest putting urban above rural since this is the order in the other tables.

The average age of the respondents was 23.3 +/– 5.3 years (the average age of women was 23.4 +/- 5.6 years, and men – 22.7 +/- 3.2 years).

Section 2.2 is well described.  It would help to have a copy of the questionnaire as an appendix (both original and a translation if possible). 

Both versions of the questionnaire are enclosed as the Additional file.

Section 2.2 is well described. It would help to also have a small table of definitions because the labels on the measures are a little ambiguous as they stand (e.g. ‘deservingness’ and ‘discarded’ can be misread). A clear description of what they mean and also perhaps how they relate to the categories that are used in other literature would help (especially since Ward’s AVRS is used as the basis).

In chapter 2.2. In section ‘Procedures and data analyzes’, the necessary clarifications are provided. The table has been dropped. Description in the text is provided.

The greater the number of points, the lower the overall level of empathy towards the rape victim (total score), the more likely, according to the respondent, that rape victim "deserved" the experience of rape (measure is the sum of points: 6, 9, 11, 13, 24, 25); the less reliable according to the respondent, that the situation that took place was actually a rape (victim credibility, measure is the sum of points: 2, 8, 14, 16, 17, 21, 22R); the more the victim is complicit for the rape (victim blame, measure is the sum of points: 3R, 5R, 7R, 10R, 12R, 15R, 19R), the more the victim deserves social exclusion because of the experience of rape and talking about this experience (victim discarded measure is the sum of points: 1, 4, 18, 20, 23). The victim deservingness, victim credibility and victim blame subscales differ in that the victim credibility scale assesses the level of belief that the situation actually occurred was rape, and the victim deservingness and victim blame subscales are based on the top-down assumption that rape took place. A high score on the victim credibility scale means a tendency to doubt whether the situation can be described as rape. The difference between the victim blame and victim deservingness scales is that a high victim blame score implies the assumption of the victim's complicity for the rape situation, i.e. some behavior or attitude that resulted in the rape. On the other hand, a high score on the victim deservingness scale means that the respondent assumes that the victim deserved a given situation - it is not necessarily about active participation in initiating a given situation, but rather that the experience of rape should not be the basis for feeling hurt, because the victim will feel hurt, e.g. it "should have been" or it was not a big hurt [13].

Lines 132-133: agree that it is reasonable to expect this to be unexpected on ethical grounds but in practice this is unlikely – generally I think the literature repeatedly shows that religious people tend to be more judgmental and less empathetic, especially as the degree of fundamentalism increases. The work of Sheldon and Parent and of Carr is cited (lines 290-300).

I agree with this observation. The following sentence was added to the discussion: The literature shows that religious people tend to be more judgmental and less empathetic, especially as the degree of fundamentalism increases.

An interaction between rurality and religiosity might be expected.  It may well be that the driver is the degree of authoritarianism rather than the ethics, since this would explain both lack of empathy among the religious and among the secular authoritarian (suggested by Carr – and an important dimension) – but I do realise this was not a dimension collected – may be a topic for further research. Certainly, the findings (lines 152-156 for example) show this is clear with respect to religion.  It does look as though this is a topic worth further work because of the finding by gender (lines 244-246) is of considerable interest.

The observations presented have been included in the chapter: Defining the direction of future research. Future research should consider other factors differentiating the dimension of attitudes towards rape victims, such as: socio-cultural differences between the urban and rural environment, the degree of authoritarianism, racism. The use of the linear regression model makes it possible to analyze the influence of gender or religiosity with the above-mentioned differentiating factors among the respondents. Additionally, it is advisable to take into account whether the victim status: refugee, different nationality, war victim differentiate the dimension of attitudes towards rape victims.

One thought that does occur (lines 134-138) is whether the difference between urban and rural might be at least in part explained by differences in social class? To what extent is the reduction (line 136-7) in differentiation of rural and urban social environments the result of growing gentrification of the areas relatively close top the city?  It might be good to cite some research in support (not suggesting it is wrong – quite the opposite – but interesting) – one feature of the urban/rural dichotomy is that there has tended to be an embedding of historical attitudes intergenerationally but which is modified by population mobility (the students have moved to an urban environment but they will still carry with them a level of historical views that may dilute as they move through the years of study).

You can read about the blurring of social and economic differences in Poland between the rural and urban environment in a very detailed report. Fedyszak-Radziejowska, B. Socio-economic situation, attitudes and values ​​of rural residents. In: Polish countryside 2020. Report on the state of the countryside. (Ed. Wilkin, J .; HaÅ‚asiewicz, A.)., Warsaw: SCHOLAR, 2020. Pp. 57–75. Available online: https://sir.cdr.gov.pl/wp-content/uploads/2020/06/Raport-o-stanie-wsi-Polska-Wies-2020.pdf (accessed on 8 May 2022). This item of literature has been supplemented. However, I have not found a satisfactory example of cultural gentrification in the urban and rural environment. The observations presented have been included in the chapter: Defining the direction of future research. Future research should consider other factors differentiating the dimension of attitudes towards rape victims, such as: socio-cultural differences between the urban and rural environment, the degree of authoritarianism, racism. The use of the linear regression model makes it possible to analyze the influence of gender or religiosity with the above-mentioned differentiating factors among the respondents. Additionally, it is advisable to take into account whether the victim status: refugee, different nationality, war victim differentiate the dimension of attitudes towards rape victims.

Table 2, 3, 4 are very useful. Just a tiny point: there seems to be a formatting slip in tables 3 and 4 (first category is bolded and underscored for columns 1-3)

Lines 169-173 are still referring to table 3 – should this precede table 4 to keep the women together?  This will, of course, make the analysis of men stand out as a bit thin, but the paragraph explains that well.

The indicated fragment (lines 169-173) has been transferred above Table 4.

Excellent discussion section. Be carefully of stereotyping (lines 273-275) – line 272-3 makes a vitally important point.  Just that ‘Asian’ is a very broad, highly diverse, grouping of ethnicities and the cultural traditions are extraordinarily diverse – although the point about under-reporting due to social consequences is well made for many of the cultures in question.

I agree. The discussion was supplemented: Asian is a very broad, highly diverse, grouping of ethnicities and the cultural traditions are extraordinarily diverse.

Lines 281-4: again, a very pertinent point. Does this study investigate differing attitudes depending on sex of the victim?  The way the text flows from line 281 implies that this information was collected – if not then perhaps prefix the sentence in line 284 with “Anderson and Quinn found that … [17]”

The sentence was redrafted: Anderson and Quinn found that, male victims were viewed more negatively than female victims [17].

Conclusion: thank you for a conclusion that is actually a conclusion – very good suggestions for further study, very good recognition of limitations.  If the opportunity arises for further study it would be interesting to add in attitudes to rape when the victim is a refugee or is foreign born.  This may show deep seated racist or religious attitudes which could mean not all victims would be treated equally (just as a topical example – would attitudes vary if the victim was Syrian or Ukrainian? – would the characteristics of the perpetrator influence the attitudes towards the victim? … touched on in this paper quite correctly – or if the person was a victim of the clergy [a thought provoked by the differences in attitudes by religiosity]).

The observations presented have been included in the section ‘Defining the direction of future research’. Future research should consider other factors differentiating the dimension of attitudes towards rape victims, such as: socio-cultural differences between the urban and rural environment, the degree of authoritarianism, racism. The use of the linear regression model makes it possible to analyze the influence of gender or religiosity with the above-mentioned differentiating factors among the respondents. Additionally, it is advisable to take into account whether the victim status, e.g. refugee, different nationality, war victim, differentiate the dimension of attitudes towards rape victims.

On behalf of all the authors, I would like to thank the Reviewer for his/her suggestions and proposed changes. Their implementation will certainly contribute to increasing the scientific value of the reviewed article.

Reviewer 3 Report

Thank you for the opportunity to review your article. I most appreciated your recognition of the limitations of your study design, particularly the self-report assessment. I found the design of your article interesting, in that your intro was much shorter than your discussion. I think it may be helpful to set up your article by moving some of what you discuss in the discussion section to your introduction section. When I was reading, it did not seem to flow as much for me from your discussion of rape statistics going to why you chose Polish medical students.

I am not personally familiar with the Attitudes toward Rape Victims Scale (ARVS). It would be helpful to have some examples of items for each of the subscales. I also would like to see some sort of reference to gauge how the scores you found in your study compare to ranges on the ARVS. In other words, I do not know if there are cut-offs for overall positive views towards rape victims versus neutral views versus negative views. As I was reading your results and discussion, it appeared as if you were framing the scores in comparison with each other and calling them "more positive," etc. Then in your conclusion, you stated that "the surveyed medical students indicates that they do not exhibit too positive attitudes towards rape victims." Where did that come from? It would seem to me that average scores in the 50s range is on the lower end of a measure that ranges from 25 to 125. This needs more clarification.

Author Response

Response to the Reviewer 3 comments:

Our responses to comments and suggestions made by the Reviewer and changes we implemented in the text are marked in red.

I found the design of your article interesting, in that your intro was much shorter than your discussion. I think it may be helpful to set up your article by moving some of what you discuss in the discussion section to your introduction section.

We complete the introduction section. The introduction now includes the definition of an attitude.  The authors of this article use the definition of attitude used in psychology and medicine, according to which the attitude is a set of emotions, beliefs, and behaviors toward a particular object, person, thing, or event. Attitudes are often the result of experience or upbringing, and they can have a powerful influence over behavior. As is known, attitudes often are enduring, they can also change under influence of certain factors such as personal experiences or persuasion of significant persons. (Major,  M.; Ulman, P. Charakterystyka wybranych postaw spoÅ‚ecznych w Polsce. Analiza statystyczna w Polsce. Zeszyty Naukowe Uniwersytetu Ekonomicznego w Krakowie, 2011, 847, 5-23. Available online: https://r.uek.krakow.pl/bitstream/123456789/1871/1/171189287.pdf , accessed on 8 May 2022). Nie zdecydowano siÄ™ na przeredagowanie dyskusji, ze wzglÄ™du na uwagi pozostaÅ‚ych 2 recenzentów.

When I was reading, it did not seem to flow as much for me from your discussion of rape statistics going to why you chose Polish medical students.

The explanation is included in the introduction.

The present study is aimed at examination of attitudes towards rape victims in a large group of Polish medical students. The authors assume that it is an important, however under-researched problem, especially when we consider the fact that medical personnel plays a crucial role in therapy of rape victims. Victims are dependent upon physicians and other medical personnel to gather medical legal evidence if the rape is to be reported and investigated. Moreover, whether they report or not, victims are dependent upon physicians for evaluation and treatment of medical problems which result from sexual assaults. Unfortunately, medical schools often did not adequately train their students in rape examination procedures and attitudes towards this special group of patients. The importance of this study is also due to the fact that so far in Poland, unlike other countries, no research has been conducted on the attitudes of medical students towards the rape victims.

I am not personally familiar with the Attitudes toward Rape Victims Scale (ARVS). It would be helpful to have some examples of items for each of the subscales.

Full versions of the questionnaire, in both languages, are enclosed as the Additional file.

I also would like to see some sort of reference to gauge how the scores you found in your study compare to ranges on the ARVS. In other words, I do not know if there are cut-offs for overall positive views towards rape victims versus neutral views versus negative views.

There are no cut-off points to distinguish between positive or negative attitudes towards rape victims. When interpreting the results, we relied on the recommendations of the authors of the ARVS scale.

In your conclusion, you stated that "the surveyed medical students indicates that they do not exhibit too positive attitudes towards rape victims." Where did that come from?

The conclusions were redrafted in the summary, as follows: ‘The surveyed medical students presented moderately positive attitudes towards rape victims’.

In the discussion, the term moderately positive attitudes towards rape victims was justified. It was found that if the average value of the total score for the entire studied group is 53.9, it is below the average value characterizing the scale, and above the value corresponding to the lower quartile, and it cannot be unequivocally determined that the attitude is positive, therefore it was assessed as moderately positive.

It would seem to me that average scores in the 50s range is on the lower end of a measure that ranges from 25 to 125. This needs more clarification.

I agree with the observation. It may range from 25 to 125 points, the average score is 75 points, the lower quartile – 50 points, the upper quartile – 100 points (the information has been supplemented in the chapter 2.2. Procedures and data analyses.)  The total score for the entire study population (1183 medical students) remains at the average level (xÌ…=53.9) and is below the average score characterizing the scale, and higher than the lower quartile (the information has been supplemented in the chapter 3. Results).

On behalf of all the authors, I would like to thank the Reviewer for his/her suggestions and proposed changes. Their implementation will certainly contribute to increasing the scientific value of the reviewed article.
